# Maternal and environmental Impact assessment on Neurodevelopment in Early childhood years (MINE): a prospective cohort study protocol from a low, middle-income country

Zoya Surani ,[1] Sadia Parkar,[1] Gul Afshan,[1] Kinza Naseem Elahi,[1] Zahra Hoodbhoy ,[1] Kiran Hilal,[2] Sidra Kaleem Jafri[1]

¹Department of Paediatrics and Child Health, The Aga Khan University, Karachi, Pakistan
²Department of Radiology, The Aga Khan University, Karachi, Pakistan

**Correspondence to**
Sidra Kaleem Jafri;
sidra.kaleem@aku.edu

## ABSTRACT

**Introduction** Environmental and psychosocial adversities negatively impact children's developmental outcomes. When these factors are experienced in early childhood—a sensitive period of development—the developing brain can be altered. While these associations have been drawn in high-income countries, it is necessary to understand child growth, neurodevelopment, and the role of environmental factors in developmental trajectories in low-income settings. The objective of this study is to longitudinally assess how demographic factors, maternal health, maternal development, and child health, are related to child development on a behavioural, cognitive, and neuroimaging level in low-socioeconomic communities.

**Methods and analysis** Mother–child dyads will be identified in the peri-urban field sites of Rehri Goth and Ibrahim Hyderi, Karachi, Pakistan. Dyads will undergo yearly assessments for 4 years beginning when the child is 1 month, 3 months or 6 months of age (+≤30 days of age) (depending on group assignment). Maternal assessments include anthropometry, behavioural, cognitive, and developmental assessments (Edinburgh Postnatal Depression Scale; Parenting Stress Index; Maternal Autonomy Index; Hurt, Insult, Threaten, Scream Tool; Reynolds Intellectual Assessment Scales (RIAS)), and biological samples collection (breast milk, blood, stool, hair). Children's assessments include anthropometry, developmental assessments (Global Scales for Early Development (GSED); RIAS), MRI brain assessments, and biological sample collection (blood, stool, hair). Using cross-sectional and longitudinal data with statistical analysis tools, associations will be quantified between brain structure (MRI) and connectivity (resting state connectivity and diffusion tensor imaging), general cognitive skills (RIAS, GSED) and environmental influences (nutrition via biological samples, maternal mental health via questionnaires) through repeated measures analysis of variance tests and $\chi^2$ tests. Quantile regression and cortical analyses will be conducted to understand how demographic factors are related to the associations found.

**Ethics and dissemination** The study has received ethical approval from the Aga Khan University Ethics Review Committee. The study's findings will be disseminated through scientific publications and project summaries for the participants.

---

**STRENGTHS AND LIMITATIONS OF THIS STUDY**

⇒ This is a longitudinal study that will assess various behavioural, cognitive, biological, and neuroimaging factors related to child development yearly for 4 years.
⇒ There will not be a direct control group in this study.
⇒ Though the sample size was calculated taking into account attrition that may occur over the study's timeline, a formal sample size calculation was not completed to determine the power of the study.

---

## INTRODUCTION

Worldwide, an estimated 250 million children under the age of 5 years fail to meet their developmental potential due to the exposure to early life adversities, such as malnutrition, psychosocial and environmental stress, and neglect.[1] Many of these children live in low-income and middle-income countries (LMICs), where there is often a profound confluence of negative environmental and psychosocial factors.[2] The first 1000 days of a child's life are widely recognised as a sensitive period of development, where the structural and functional architecture of the brain is laid, and lifelong patterns of physical, cognitive, and socioemotional health are established.[3]

Throughout this period, the fetal and infant's brain increases almost threefold in volume, and its extensive neuroarchitecture matures through processes including neurogenesis, migration, synaptogenesis, and myelination.[4] This transformation is fueled by the mother and child's nutritional and energy resources, and shaped by cascades

BMJ

of genetic and environmental interactions—modulated through psychosocial and caregiving relationships.[3 5] Previous neuroimaging studies provide clear evidence of how these factors can alter the developing brain, particularly when these adversities are experienced during the early life years.[6]

Infants are inherently competent in their ability to initiate relationships, explore, seek meaning and learn, but are also vulnerable and depend entirely on caregivers for their survival, emotional security, modelling of behaviours, and to teach them the norms of the physical and sociocultural world that they inhabit.[7] This sensitive period, then, is a double-edged sword. While positive and enriching environments can promote healthy brain development, neglect and abuse can impair maturing brain systems and disrupt cognitive and behavioural outcomes.[8 9] Early adversities disrupt normative developmental trajectories with lifelong consequences, including impaired intellectual functioning, depression, and other mental health disorders.[6 10] Therefore, adversity and resource inequality not only contribute to poor brain development but are also an outcome of it, potentially driving a vicious intergenerational cycle.[11]

While existing literature on this topic has largely focused on understanding the neurodevelopmental impact of the environmental factors as they exist in high-income areas, literature is evolving to include more data from LMICs—as understanding the nature of adversities in different environments is an important consideration, particularly in LMICs where the burden to do so is greater. Of the studies that aim to understand neurodevelopment, there has been a recent focus on using MRI as the preferred modality of neuroimaging due to its anatomical tissue contract, ability to visualise anatomical structure and physiological function, and its safety in use.[12] By using MRI to image young children, there is a greater ability to visualise and understand brain plasticity, and development of perceptual and cognitive skills.[13]

The two major studies conducted on this topic in LMICs were completed in Bangladesh and India. In the Bangladesh Early Adversity Neuroimaging study, associations between psychosocial risks (maternal distress, caregiving experiences), biological risks (growth, inflammation), developmental outcomes (Mullen Scales, Wechsler Preschool and Primary Scale, Bayley Scales), and neurodevelopment (electroencephalography, MRI) were assessed in children aged 6 months to 7 years.[14] In this study, as well as another conducted in India to understand neurodevelopmental outcomes in infants up to 18 months of age, researchers found that adverse factors present in LMIC environments are associated with outcomes on the cognitive and neural level.[14 15] These studies highlight the importance of using MRI in LMICs to investigate the impact of these environments.

Despite the presence of studies in LMICs aimed at explaining children's neurodevelopment in these environments, there is a decreased understanding of the mechanisms involved in how and why children's outcomes are impacted. These are important considerations because before effective interventions can be designed to minimise adversities and their impact on children's development and neurodevelopment, it is necessary to understand the role of environmental factors. Knowing that the early years are a time of immense brain development and extremely susceptible to factors in the environment, it is necessary to comprehensively map children's growth and development—assessing both environmental factors and brain structure continuously in low-resource environments.

Pakistan is among the countries with the highest rates of malnutrition and its progress in perinatal, neonatal and child health has been much slower than in other South Asian countries.[16 17] Karachi, the largest metropolitan city in Pakistan and 12th largest city in the world with a linguistically, ethnically, and religiously diverse population is an excellent surrogate to study the impact of the environment on a child's development. The primary objective of the current study is to longitudinally map trajectories of child growth and neurodevelopment from early infancy (1 month) to 3.5 years of age using longitudinal MRI scanning and neurodevelopmental assessments in Karachi, Pakistan. The secondary objective of this study is to investigate the relationships between modifiable environmental factors (eg, early child nutrition, maternal stress) with evolving brain structure, brain function, and maturing cognitive skills. We hypothesise that children with exposure to conditions like maternal and child malnutrition and adverse childhood experiences will have poor development with macro-level structural brain differences in comparison to children who were not exposed to these conditions.

## METHODS AND ANALYSIS
### Setting
In Karachi, to facilitate the large number of patients per day, The Aga Khan University (AKU) Hospital has created a massive network with 4 secondary care hospitals, 30+ medical centres, 290+ clinical laboratories and 20+ pharmacies in over 120 cities, and also provides home healthcare services across Pakistan. AKU Hospital in Karachi is connected with many primary healthcare clinics (PHCs) in the peri-urban coastal regions of the city and has a 3 Tesla (3T) MRI machine available on-site. The PHCs are maintained by the Department of Paediatrics and Child Health at AKU, and have quarterly household surveillance for key maternal and child health indicators along with antenatal care, immunisation, and physician services for children under the age of 5 years. For these reasons, the provision of services at PHCs connected with the larger tertiary hospital make this area an ideal setting for this study. The study will be conducted at two of these PHCs, Rehri Goth and Ibrahim Hyderi, which have been operating in the communities for over a decade.

## Design

This study is a longitudinal cohort study and will be built on the Pregnancy Risk Stratification Innovation and Measurement Alliance (PRiSMA) study (ERC Number: 2021-5920-15518). The PRiSMA study is a longitudinal study that enrols pregnant women within the first 20 weeks of gestation and follows them and their newborns until 1 year after delivery. The current study will map trajectories of child growth and neurodevelopment from 1 month to 3.5 years of age. Additionally, the reporting of this study conforms to the Strengthening the Reporting of Observational Studies in Epidemiology statement.[18]

## Duration

The study will be conducted over a 3.5-year period (May 2022 through December 2026).

## Inclusion and exclusion criteria

The inclusion criteria are:
► Singleton infants aged 1 month +≤30 days of age, 3 months +≤30 days of age, or 6 months +≤30 days of age and their mothers.

The exclusion criteria are:
► Infants with congenital anomalies (established on clinical visits).
► Infants with history of hypoxic-ischaemic insult, delayed cry, requiring oxygen, or hospitalisation.
► Maternal usage of antidepressive or anti-epileptic medications during pregnancy.
► Fetal exposure to maternal smoking (cigarettes).
► Ultrasound abnormalities (eg, ventriculomegaly, renal agenesis, gastrointestinal structural anomalies).
► Conditions like pre-eclampsia, high blood pressure, or diabetes mellitus in pregnancy.

## Sample size calculation

A formal sample size calculation to obtain the power of the study was not calculated. We aimed to collect data on 150 mother–child dyad pairs over the study duration, basing our sample size on what was found feasible in previous neuroimaging infant studies.[14 15] However, keeping attrition of 40% due to identified challenges (eg, difficulty to obtain follow-up data due to family relocation, family refusal), approximately 250 mother–child dyads will be recruited.

## Recruitment procedure

A list of households with newborns will be shared by the PRiSMA study team on a monthly basis. After households with infants of appropriate age are identified, mother–child dyads will be approached via simple random sampling to reduce any sampling bias. At the mother's home, research staff will ask questions related to their and their child's medical history to determine their eligibility (based on the exclusion criteria).

Once written informed consent is provided by the parent/guardian, the enrolment visit will be performed, and a unique study participant identification number (ID) will be assigned to the infant and mother, where they will both have the same ID number, but labelled with 'infant' or 'mother' to distinguish them (eg, infant ID: Infant-1234, mother ID: Mother-1234). The participants will be recruited in three groups: Group 1 will include 100 mother–infant dyads with infants that are 1 month +≤30 days of age, Group 2 will include 100 mother–infant dyads with infants that are 3 months +≤30 days of age, and Group 3 will include 50 mother–infant dyads with infants that are 6 months +≤30 days of age. As neuroimaging scans in natural sleep are more likely to be successful in infants of younger age and MRI scans with sedation (if needed) may be more challenging to obtain enrolments for, the group sizes of the younger groups (1 month and 3 months) are larger than that of the older infants (6 months). This split will also allow the team to obtain images and samples at different time points during the early childhood period without the need for repetitive sampling of any one mother–child dyad.

The study involves the mother–child dyads completing 4 yearly visits, beginning when the child is an infant (1 month/3 months/6 months) until they are 3 years and 1 month/3 months/6 months of age. Each visit will comprise of the child's MRI brain scan, followed by the collection of information and biological samples. For the MRI scan, the family will be transported to AKU, accompanied by members of the research team. Once the scan is completed, estimated to take no longer than 1 hour (prescanning (5 min), child placement (10 min), scanning time (30+ min, depending on repeated measures)), the demographic information, maternal physical and mental health assessments, child's history, child physical health and developmental assessments, and biological samples will be collected on the following day at the PHC at the study site. The collection of the assessments and samples will take approximately 2 hours. The collection procedure of non-MRI data will be adapted to ensure effective use of resources once the MRI scan is successfully performed. Additionally, quality control programmes are in place for data collection to ensure there are no changes in the measurement of variables over the study's duration. Weekly quality checks will be conducted under the supervision of master trainers for all behavioural, cognitive, and developmental assessment tools used in the study. For the neuroimaging data, to ensure the scanner is operating correctly and producing reliable images, radiologists conduct routine quality control and assurance procedures weekly, such as system checks and calibrations (with a phantom), under the supervision of the biomedical team. In addition, a service engineer performs thorough diagnostic/maintenance tests on a quarterly basis to confirm the machine is working as intended. If there are any machine or software malfunctions, scans will not be completed during that time. Lastly, the quality of biological samples will be checked by the laboratory team and frequent quality management of all collected data will be done by the Data Management Unit.

**Figure 1** Summary of measures collected in each of the four visits from both the mother and child. *MRI brain scanning will occur at Aga Khan University main campus. All other measurements will be collected at the Rehri Goth and Ibrahim Hyderi field sites. RBC, red blood cell; TIBC, total iron-binding capacity; TSH, thyroid stimulating hormone.

A summary of the visits is provided in figure 1 and descriptions of each measure to be collected are provided in the next section.

## Measures of outcomes

1. Demographics and history
   – Demographic/socioeconomic information: Demographic and socioeconomic data will be collected at all four visits. The data will include detailed maternal pregnancy history (gravidity, live births, pregnancy spacing, maternal age at each pregnancy), family composition (child ages and sex), socioeconomic and household asset information (mother and father education and occupation, religion, caste, household income, home size and number of rooms, toilet type, electricity access, number of mobile phones, housing type, cooking fuel, home, land and/or animal ownership, and transportation access).
   – Child's history: A detailed assessment regarding the child's well-being, comorbidities if any, major hospitalisations, child diarrhoea events, antibiotic use, and vaccination history will be collected at all visits. Labour and delivery information, as well as birth anthropometry, will be obtained from the PRiSMA records.

2. Maternal assessments
   – Physical health assessments:
     i. Anthropometric measurements: Assessment of maternal anthropometry, that is, weight and height, will be recorded using a Calibrated Weighing Scale (Seca 874 Mobile Flat Scale) and measuring tape (Seca 212 Measuring Tape), respectively, at all visits.
     ii. Biological sample collection: Laboratory investigations of serum iron, total iron-binding capacity (TIBC), red blood cell (RBC) folate, vitamin B-12 levels, vitamins A and D, blood lead levels, and thyroid stimulating hormone (TSH) will be done at visit 1. A sample of the mother's hair will be collected at visits 1 and 3 to assess for cortisol. A breastmilk sample (20 mL) will be collected from the mother at visit 1 and the composition (general macronutrient (protein, fat, and carbohydrate content), as well as cholesterol, choline, and vitamin B content) will be determined. Stool sample for faecal microbiome analysis will be collected from the mother at visit 1.
   – Behavioural and cognitive assessments: Behavioural and cognitive assessments will be administered at all visits.
     i. Maternal mental health: Will be assessed using Urdu translations of the Edinburgh Postnatal Depression Scale (EPDS),[19] Parenting Stress Index (PSI) long-form,[20] and through collecting the hours of maternal rest per day. The EPDS and PSI have been locally validated.[21–23]

ii. Maternal autonomy, family violence, addiction and neglect: Will be assessed using Urdu-translated versions of the Maternal Autonomy Index,[24] and the Hurt, Insult, Threaten and Scream Tool,[25] which have been locally validated.[26 27]

– Developmental assessment:

The Reynolds Intellectual Assessment Scales (RIAS)[28] will be used to assess the mother's development at all visits. The RIAS measures general intelligence and non-verbal intelligence, and is administered to individuals ages 3–94. The RIAS has been locally validated.[29]

3. Child assessments

– Physical health assessments:

i. Anthropometric measurements: Weight, height, and head circumference of the child will be recorded using a Calibrated Weighing Scale (Seca 334 Mobile Digital Baby Scale), infant measuring board (Seca 416 Infantometer), and measuring tape (Seca 212 Measuring Tape), respectively. Mid-upper arm circumference will also be measured for children above 6 months of age at every visit. The WHO Z-scores will be used for anthropometric interpretation and standards for weight, length/height, weight-for-length/height, and head circumference-for-age will be calculated. Child anthropometric measurements will be collected yearly (visits 1–4) and analysed within their enrolment group and visit number.

ii. Biological sample collection: Laboratory investigations of serum iron, TIBC, RBC folate, vitamin B-12 levels, vitamins A and D, blood lead levels, and TSH will be completed yearly (visits 1–4). Stool samples will be collected yearly (visits 1–4) and will be assessed for faecal microbiome (visits 1 and 3) and helminth (visits 2, 3 and 4). Hair sample to assess cortisol levels will be collected at visits 3 and 4.

– Developmental assessments:

The Global Scales for Early Development (GSED)[30] (0–6 years) and RIAS (non-verbal) (3–94 years) tools will be used to assess the child's development. The GSED will be administered at all four visits, while the RIAS will only be administered to the child at visit 4 (when the child is >3 years old).

i. GSED: The WHO has developed and is currently validating the GSED, which can be used to assess children's development in the early years of life across the globe, including in Pakistan. As part of this scale, the short-form (SF) and long-form (LF) will be used. Both SF and LF are unidimensional and provide a single age-adjusted score that represents a child's level of development: The SF uses a caregiver-reported questionnaire while the

LF is a directly administered (observed) tool assessing the child's abilities and behaviours.

ii. RIAS: The RIAS measures general intelligence and non-verbal intelligence, and is administered to individuals ages 3–94. As the participants will be >3 years old only at visit 4, the RIAS will be administered only at that visit.

– MRI brain scanning:

At each of the four visits, MRI scans (lasting no longer than 1 hour) will be conducted using a 3T MRI machine (Toshiba Titan, Canon Medical Systems) equipped with a 32-channel coil, which provides high field strength, multiphase imaging, and advanced motion correction capabilities. To optimise the functionality and performance of the Titan scanner, the specialised Ventage V.2.5 software will be used.

The sequences that will be performed as part of analysis for this study include diffusion tensor imaging (DTI), T1-weighted MPRAGE and T2 volume anatomical imaging, and functional MRI (fMRI). Information about the key parameters and sequence timing are included in table 1. As part of the hospital protocol, fluid attenuated inversion recovery and Flow Sensitive Black Blood susceptibility-weighted imaging will also be performed and reviewed by a physician, but these sequences will solely be used for clinical information. The parameters for these sequences are included in online supplemental information, table 1.

For MRI brain scanning, the mother and child will be brought to AKU accompanied with members of the research team. Once the child appears to be in deep sleep (15–20 min after the child went to sleep), the MRI will be performed. The child will be secured in an immobiliser (MedVac Immobilisation Bag, Infant Splint; MRI Med, Petaluma, California, USA) and transferred from their crib or bed to an MRI-compatible cart. The immobiliser will be used to reduce subtle body movement during the scan (eg, movement from deep sleep breathing). The child's head will be carefully positioned, and a removable sound-insulating foam insert (Ultra Barrier HD Composite; American Micro Industries, Chambersburg, Pennsylvania, USA) will be securely placed on the child's ears. The process will be facilitated by a research coordinator, and parents will be invited to stay with the infant to allay any anxiety. A paediatric pulse oximeter will be attached to the child's finger or toe to monitor the child during the scan. A research assistant will remain inside the scanning suite in case the child wakes up during the scan.

With children scanned in natural sleep, there is a possibility that they may wake up during the scan. If this happens, the child will be removed from the scanner and nursed and swaddled prior to continuing the scan. The sequences affected by participant motion will be repeated once the child is asleep again. The scan will ideally be performed without sedation at any stage of the scan, using strategies developed by

**Table 1** Key parameters of MRI sequences performed at all visits

| Sequence | Diffusion weighted imaging 30 axis | Functional MRI | T2 volume | Three-dimensional T1-weighted MPRAGE |
|---|---|---|---|---|
| Orientation | Axial | Axial | Sagittal | Sagittal |
| TR (ms) | 5090 | 2400 | 3200 | 1819.5 |
| TE (ms) | 82 | 25 | 352 | 3.384 |
| FA (degrees) | 90 | 90 | 90 | 12 |
| FOV (mm) | 230×230 | 240×240 | 231×231 | 160×160 |
| Matrix size | 160×160 | 96×96 | 512×512 | 512×512 |
| No. of Slices | 28 | 21 | 260 | 150 |
| Thickness (mm) | 4 | 3 | 1 | 1.48 |
| Spacing (mm) | 4 | 4 | 0.5 | 0.75 |
| Voxel size (mm$^3$) | 1.438×1.438×4 | 2.5×2.5×4 | 0.451×0.451×0.5 | 0.312×0.312×0.75 |
| Phase encoding direction | j; Anterior to Posterior (A to P) | j; A to P | i; Right to Left (R to L) | i; R to L |
| b-values (sec/mm$^2$) | 0/1000 | | | |
| Time (s) | 05:38 | 03:41 | 04:32 | 09:55 |

FA, Flip Angle; FOV, Field of View; TE, Echo Time; TR, Repetition Time.

Dean et al.[13] In case the child does not sleep, we will use minimally conscious sedation routinely employed in the paediatric population. For infants up to 3 months, this includes chloral hydrate, 25 mg per kg given orally, while for children 6 months and above, procedural sedation will be used which will be carried out by a credentialed physician and sedation nurse. The sedation medication will be titrated according to the child's weight, and children requiring sedation will be admitted for pre, intra, and post sedation monitoring.

During the follow-up visits, though the imaging data may be difficult to acquire in unsedated toddlers, the scans will be initially attempted in natural sleep. The data will be carefully reviewed to exclude low quality data and data that exceeds our motion threshold. Scans with sedation will be used as a last resort and will be performed rarely due to possible risks that might occur with annual evaluations with sedation.

### Patient and public involvement

Patients and the public were not involved in the design of this study. However, study participants will be informed regarding the study findings after the study results are published and their help will be enlisted to help with the dissemination of research findings.

### Statistical analysis plan

Analysis of participant characteristics will be conducted using Stata V.17. Participant characteristics will be described using means and SD for continuous variables (anthropometric measurements, maternal behavioural, cognitive, and developmental assessments scoring, and child developmental assessments scoring) and percentages for categorical variables (biological samples, demographics and child's history information).

MRI brain analysis will be performed locally, as experts from the funding agency will train the local Data Management Unit team at AKU. MRI data will be quality checked by a radiologist and radiologist technician to ensure there were no abnormalities found, all sequences were completed, and there was limited motion. Visualisation of data will be done using Mango and Horos tools. Analysis of neuroimaging data will be completed using the Flywheel platform and the embedded FreeSurfer, FSL, and MRtrix3 analysis tools. Data will be segmented, skull-stripped, scaled to standard space, and registered to age-matched templates that will be created locally and matched with existing templates (eg, from the National Institute of Health). For structural data, estimates of total brain tissue volume, thickness, and surface area will be acquired. FMRI data will be preprocessed and motion corrected to analyse differences in resting-state activity (as the scans will occur in natural or sedated sleep). Fractional anisotropy values will be obtained from DTI to understand the shape of diffusion and will be visualised with tractography.

Ultimately, patterns of structural and functional development will be characterised, and associations will be conducted between cognitive outcomes and deviations associated with birth outcomes and environmental factors at various time points. Imaging data will be paired with neurocognitive assessments (GSED, RIAS, etc) performed at predetermined intervals (GSED at visits 1–4, RIAS at visit 4). We will quantify the associations (eg, with a $\chi^2$ test) between maternal and child health (via levels of micronutrients in the blood and breast milk) on measures of fetal and infant growth and neurodevelopment. We will use repeated measures analysis of variance tests to

determine statistically significant differences between measures collected at various time points–child anthropometric measurements, biological samples, child developmental assessment scores, and child MRI scanning via volume estimates through structural MRI, resting-state connectivity patterns through fMRI, and functional connectivity through DTI. Quantile regression analyses will be performed to understand if demographic factors (eg, socioeconomic status) and/or child sex are related to any of the associations found between cognitive skills, environmental influences, and emerging neurobehaviour. Cortical thickness analyses will be conducted to specifically understand the correlation between demographic factors and structural differences (via cortical thickness as found through T1-weighted MRI).

As this is a longitudinal study, due to identified challenges that may result in a loss of follow-up, we have kept attrition at 40%, thus increasing our participant recruitment to 250 mother–child dyads for our ultimate sample size of 150 mother–child dyads. Additionally, during the analysis, if there is missing data, the data will be addressed appropriately to ensure the remaining data are able to be maximally used.

## ETHICS AND DISSEMINATION

Ethics approval of the study was obtained from the Ethics Review Committee of AKU (Reference Number: 2022-7181-21831). Written informed consent will be voluntarily obtained from all participants, and participants will be informed of their complete right of refusal to initiate or withdraw at any stage of the study with no obligations. Consent for the biological samples will be obtained as part of the consent for participation in the study. Hospital data transfer policy will be employed, and all data collected and/or shared from participants will be de-identified.

The study's findings will be disseminated through scientific publications and medical education seminars. Participants will be informed of the study's findings through project summaries with the key findings included. As this study may provide insight to factors in the environment that will influence child development, the key points from the study will be shared with stakeholders and policymakers (eg, Pakistan Pediatrics Association, Government of Sindh), who can use the study's findings to plan interventions for the physical and mental health of mothers and children at national, regional, and international levels.

**Acknowledgements** We thank our Aga Khan University Senior Research Assistants Sahiba Inayat, Adeeba Samreen, Shumaila Samito, our data collectors and field team for their help in project management and development.

**Contributors** ZS, SP, GA, KNE, KH, ZH and SKJ were involved in the concept, grant writing protocol development and intellectual inputs for this project. SKJ, ZH and KH provided expert advice and technical training. All authors contributed towards the development of this manuscript.

**Funding** This work was supported by The Bill & Melinda Gates Foundation (grant number: INV-004939).

**Competing interests** None declared.

**Patient and public involvement** Patients and/or the public were involved in the design, or conduct, or reporting, or dissemination plans of this research. Refer to the Methods section for further details.

**Patient consent for publication** Not applicable.

**Provenance and peer review** Not commissioned; externally peer reviewed.

**ORCID iDs**
Zoya Surani http://orcid.org/0000-0002-5384-5643
Zahra Hoodbhoy http://orcid.org/0000-0002-0439-8293

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
