## [Reviewer comments · BMJ Open]

ARTICLE DETAILS

TITLE (PROVISIONAL)	Maternal and Environmental Impact Assessment on Neurodevelopment in Early Childhood Years (MINE): A Prospective Cohort Study Protocol from a Low, Middle-Income Country
AUTHORS	Surani, Zoya; Parkar, Sadia; Afshan, Gul; Elahi, Kinza; Hoodbhoy, Zahra; Hilal, Kiran; Jafri, Sidra Kaleem

VERSION 1 – REVIEW

REVIEWER	Kim Cecil University of Cincinnati College of Medicine
REVIEW RETURNED	24-Mar-2023

GENERAL COMMENTS	It is very nice to see a study such as this has been funded and initiated. Biological Samples: Storage of urine specimens from mother and child would be useful for future analyses of environmental exposures to second-hand tobacco smoke (cotinine), pesticides and other man-made environmental chemicals. Protocol: The expected duration for each of the components of the protocol should be presented. The portion of greatest concern is the length of the individual MRI examination. MRI Protocol: The manuscript details a very ambitious MRI protocol that requires a state of the art MRI system with high performance gradients that routinely undergoes quality assurance testing. The manuscript should be specified with greater technical detail about MRI acquisition and quality assurance procedures. While infants can be nursed/fed, swaddled and then positioned into the scanner during sleep for imaging data collection, the protocol will be very difficult to fully complete in unsedated toddlers. The sounds of the MRI gradients during the echo planar imaging sequences, especially for diffusion and functional MRI often wake these young children. There will need to be careful review of the data to exclude low quality and data with motion. There should be an expectation of a large amount of missing data. An annual research MRI evaluation with sedation poses elevated risks to the individual child. This is an ethical concern. Also, the manuscript indicated the MRI brain analyses will be performed at Brown University. These also need to be specified. Statistical Plan: Needs further description especially in regards to neuroimaging. Citations: Across the manuscript, there are more recent citations relevant to this work available.
--

REVIEWER	Peter Etim Ekanem Mekelle University, Human Anatomy
REVIEW RETURNED	29-Mar-2023

GENERAL COMMENTS	I commend the authors for their initiative to carry out this work. It would sure provide good data and references to similar and related work in other countries and economy. Their concept, methodology and expected outcomes would also shade light into the factors that are averse to the development of a child taking the mother -child relationship into consideration. Thank you for a great initiative. However, I have observed the following.:  1. The title should be reviewed with phrases that should include the scope of the work as the present title seems to suggest MRI studies only, 2. The significance of the study is limited but should include stakeholders and policy makers who are able to take action on the findings. 3. Apart from the location where the facilities to be used for the research are found, the background lacks other rationales for choosing Karachi, Pakistan as the location for this work. 4. page 5 line 53; considering that the study involves a dyad, the study criteria should also include maternal as stated in page 6 line 23 5. page 6 line 6; Are delayed cry, requiring oxygen and hospitalization here used as a triad for hypoxic ischemic insult, which can occur postnatally as well? If so, are children with neonatal admission requiring oxygen without a delayed cry included? 6. page 6 line 16; what were the limitations behind calculation of sample size, seeing that there are similar studies conducted in India and Bangladesh to draw from as mentioned in your background? 7. page 6 line 26; Please elaborate on how the simple random sampling will be conducted. 8. page 6 line 31-37. State the rationale for selecting ages 1, 3, and 6 months and 50,100, and 100 sample distribution as per ages assigned respectively. 9. page 6 line 40-41, 50-53; it would be good to mention the frequency of these visits and the MRI scans in the methodology as opposed to leaving that in the figure alone 10. page 6 line 48; Elaborate on which quality control measures have been put in place for data collection as mentioned, to ensure there are no changes in the measurement of variables over the duration of the study. 11. The information in the figure given should match with the methodology as much as possible, especially in key areas. Example, visit 2 in the figure states demographic information alone yet the methodology states demographic and socioeconomic data (page 7 line 19) 12. Maternal factors in page 7 line 36-39 are specified for assessment at visit 1. However, maternal factors like anthropometry and behavioral and cognitive (page 8 line 9) outcomes can change at any point of a child's growth, especially in regard to the ages included in the study during follow up, thus affecting neurodevelopment. These maternal factors could be assessed at every visit. 13. Child's history, page 7 line 27 is specified at the first year, yet this history is contributory at every stage of the early child development. Major hospitalizations due to CNS or severe infections, for example, at any point in the first year or two of life can affect an otherwise previously well child's neurodevelopment and MRI finding. It is recommended that this history be taken at every
---

	visit. 14. page 8 line 23-28; Specify ages in which MUAC will be conducted as opposed to all the children stated in the work. 15. page 8 line 23-28 specify which growth curve will be used for anthropometric interpretation.
--	---

VERSION 1 – AUTHOR RESPONSE

Reviewer 1	
Biological Samples: Storage of urine specimens from mother and child would be useful for future analyses of environmental exposures to second-hand tobacco smoke (cotinine), pesticides and other man-made environmental chemicals.	Thank you for this suggestion - we reached out to labs to see what options might be possible for us in the follow-up visits. We are currently exploring options related to checking for cotinine in keratinized matrices (hair, nails) as they have a longer retrospective exposure period than urine or blood, but we have not been able to finalize anything yet.
Protocol: The expected duration for each of the components of the protocol should be presented. The portion of greatest concern is the length of the individual MRI examination.	This has been added (page 6, lines 9-10, 13-14). Additionally, a scan card with the exact MRI sequence timings is included in a new figure (Figure 2). Revision: Once the scan is completed (estimated to take between 30 minutes to 1 hour, depending on participant cooperation), the demographic information, maternal physical and mental health assessments, the child’s history, child physical health and developmental assessments, and biological samples will be collected on the following day at the primary health center (PHC) at the study site. The collection of the assessments and samples will take approximately two hours.
MRI Protocol: The manuscript details a very ambitious MRI protocol that requires a state of the art MRI system with high performance gradients that routinely undergoes quality assurance testing. The manuscript should be specified with greater technical detail about MRI acquisition and quality assurance procedures. While infants can be nursed/fed, swaddled and then positioned into the scanner during sleep for imaging data collection, the protocol will be very difficult to fully complete in unsedated toddlers. The sounds of the MRI gradients during the echo planar imaging sequences, especially for diffusion and functional MRI often wake these	Information regarding the MRI protocol has been updated (page 8, lines 18-22, 36-39, 45; page 9, lines 1-14). This includes a more in-depth explanation of the MRI sequences, infant scanning procedure, sedation medication, follow-up scanning, and quality check. The MRI analyses information was also updated (page 9, lines 28-37) and will be performed locally rather than with Brown University. Revision: MRI brain scanning will be completed using a 3T MRI (Toshiba) machine at every visit (1-4). The total scan time of the MRI scan is 32:27 minutes, which includes T1 and T2-weighted anatomical imaging, diffusion tensor imaging, functional MRI, and FLAIR sequences. The scan card with sequence and sequence timing is included in Figure 2. For MRI brain scanning, the mother and child will be brought to AKU accompanied with members of the research team. Once the child appears to be in deep sleep (15-20 mins after the child went to sleep), the MRI will be performed. The child will then be secured in an immobilizer (MedVac Immobilization

young children. There will need to be careful review of the data to exclude low quality and data with motion. There should be an expectation of a large amount of missing data. An annual research MRI evaluation with sedation poses elevated risks to the individual child. This is an ethical concern. Also, the manuscript indicated the MRI brain analyses will be performed at Brown University. These also need to be specified.

Bag, Infant Splint; MRI Med, Petaluma, CA, USA) and transferred from their crib or bed to an MRI-compatible cart. The immobilizer will be used to reduce subtle body movement during the scan (e.g., movement from deep sleep breathing). The child's head will be carefully positioned, and a removable sound-insulating foam insert (Ultra Barrier HD Composite; American Micro Industries, Chambersburg, PA, USA) will be securely placed on the child's ears. The process will be facilitated by a research coordinator, and parents will be invited to stay with the infant to allay any anxiety. A pediatric pulse oximeter will be attached to the child's finger or toe to monitor the child during the scan. A research assistant will remain inside the scanning suite in case the child wakes up during the scan.

Though the total scan time is 32:27 minutes, repeating sequences depending on motion artifacts may result in the scan lasting longer. With children scanned in natural sleep, there is a possibility that they may wake up during the scan. If this happens, the child will be removed from the scanner and nursed and swaddled prior to continuing the scan.

The scan will ideally be performed without sedation at any stage of the scan, using strategies developed by Dean et al (13). In case the child does not sleep, we will use minimally conscious sedation routinely employed in the pediatric population. For infants up to 3 months this includes chloral hydrate, 25mg per kg given orally, while for children 6 months and above, procedural sedation will be used which will be carried out by a credentialed physician and sedation nurse. The sedation medication will be titrated according to the child's weight and children requiring sedation will be admitted for pre, intra, and post sedation monitoring.

During the follow-up visits, though the imaging data may be difficult to acquire in unsedated toddlers, the scans will be initially attempted in natural sleep. The data will be carefully reviewed to exclude low quality data and data that exceeds our motion threshold. Scans with sedation will be used as a last resort and will be performed rarely due to possible risks that might occur with annual evaluations with sedation.

While the length of the MRI exam will be affected by various factors, including patient motion and machine problems or troubleshooting during the scan (i.e. scanner calibration), all MRI data will undergo routine quality assurance. MRI data will be checked immediately after acquisition to ensure 1) no abnormalities were found, 2) all sequences were completed, and 3) there was limited motion. The MRI machine will also be continuously checked to ensure it is running properly, and if there are any machine or software malfunctions, scans will not be completed during that time.

Statistical Plan: Needs further description especially in regards to neuroimaging.	The statistical plan was updated with in-depth explanation of neuroimaging analyses (page 9, lines 28-37). Revision: MRI brain analysis will be performed locally, as experts from the funding agency will train the local Data Management Unit team at AKU. MRI data will be quality checked using Horos and Mango visualization tools. Analysis of neuroimaging data will be done using the Flywheel platform and the embedded FreeSurfer, FSL, and MRtrix3 analysis tools. Data will be segmented, skull-stripped, scaled to standard space, and registered to age-matched templates that will be created locally and matched with existing templates (e.g., NIH). For structural data, estimates of total brain tissue volume, thickness, and surface area will be acquired. Functional MRI data will be preprocessed and motion corrected to analyze differences in resting-state activity (as the scans will occur in natural or sedated sleep). Fractional anisotropy values will be obtained from the diffusion imaging to understand the shape of diffusion and will be visualized with tractography.
Citations: Across the manuscript, there are more recent citations relevant to this work available.	The citations have been updated to include more recent sources. The citations for the tools, however, were kept the same, as these did not change. Changed citations have been marked in-text and in the references.
Reviewer 2	
The title should be reviewed with phrases that should include the scope of the work as the present title seems to suggest MRI studies only,	We have modified the title to: Methods to Image and Assess the Neurodevelopmental Impact of Environment in the Early Years (MINE): A Prospective Cohort Study Protocol from a Low, Middle-Income Country Please let us know if there is another title you find suitable.
The significance of the study is limited but should include stakeholders and policy makers who are able to take action on the findings.	This has been updated in the Ethics and Dissemination section (page 10, lines 27-30). Revision: The study's findings will be disseminated through scientific publications and medical education seminars. Participants will be informed of the study's findings through project summaries with the key findings included. As this study may provide

	insight to factors in the environment that will influence child development, the key points from the study will be shared with stakeholders and policymakers, who can use the study's findings to plan interventions for the physical and mental health of mothers and children at national, regional and international levels.
Apart from the location where the facilities to be used for the research are found, the background lacks other rationales for choosing Karachi, Pakistan as the location for this work.	We have included additional information about choosing Karachi as the site in both the Background (page 4, lines 16-20) and in the Methods, Setting (page 4, lines 32-43). Revision: Background: Pakistan is among the countries with the highest rates of malnutrition and its progress in perinatal, neonatal, and child health has been much slower than in other South Asian countries (16,17). Karachi, the largest metropolitan city in Pakistan and 12th largest city in the world with a linguistically, ethnically, and religiously diverse population is an excellent surrogate to study the impact of environment on a child's development. Methods: In Karachi, to facilitate the large number of patients per day, The Aga Khan University Hospital has created a massive network with 4 secondary care hospitals, 30+ medical centers, 290+ clinical laboratories, and 20+ pharmacies in over 120 cities, and also provides home health care services across Pakistan. Aga Khan University Hospital in Karachi is connected with many primary health care clinics (PHCs) in the peri-urban coastal regions of the city and has a 3T MRI machine available onsite. The PHCs are maintained by the Department of Paediatrics and Child Health at AKU, and have quarterly household surveillance for key maternal and child health indicators along with antenatal care, immunisation and physician services for children under the age of 5 years. For these reasons, the provision of services at PHCs connected with the larger tertiary hospital make this area an ideal setting for this study. The study will be conducted at two of these PHCs, Rehri Goth (RG) and Ibrahim Hyderi (IH), which have been operating in the communities for over a decade.
page 5 line 53; considering that the study involves a dyad, the study criteria should also include maternal as stated in page 6 line 23	This has been updated (page 5, line 13). Revision: All singleton infants aged 1 month, 3 months, or 6 months of age and their mothers
page 6 line 6; Are delayed cry, requiring oxygen and hospitalization here used as a triad for hypoxic	They are not used as a triad - any of the individual symptoms would result in exclusion. This has been updated (page 5, line

ischemic insult, which can occur postnatally as well? If so, are children with neonatal admission requiring oxygen without a delayed cry included?	16). Revision: Infants with history of hypoxic-ischemic insult (any of the three: delayed cry, requiring oxygen, or hospitalization)
page 6 line 16; what were the limitations behind calculation of sample size, seeing that there are similar studies conducted in India and Bangladesh to draw from as mentioned in your background?	In previous studies, we were unable to find the reasonings for the sample size selected, so we based our sample size on previous infant neuroimaging studies. This has been updated (page 5, lines 26-27). Revision: A formal sample size calculation to obtain the power of the study was not calculated. We aimed to collect data on 150 mother-child dyad pairs over the study duration, basing our sample size on what was found feasible in previous neuroimaging infant studies (20,21). However, keeping attrition of 40% due to identified challenges (e.g., difficulty to obtain follow-up data due to family relocation, family refusal, etc.), approximately 250 mother-child dyads will be recruited.
page 6 line 26; Please elaborate on how the simple random sampling will be conducted.	This has been updated (page 5, lines 32-36). Revision: A list of newborns will be shared by the PRiSMA study team on a monthly basis. All infants will be screened for eligibility, and eligible mothers enrolled in the PRiSMA project will be approached at their home by research staff and explained the purpose of the current study. The mother-child dyads will be chosen and approached through simple random sampling to reduce any sampling bias.
page 6 line 31-37. State the rationale for selecting ages 1, 3, and 6 months and 50,100, and 100 sample distribution as per ages assigned respectively.	This has been updated with the reasoning and revised sample distribution (page 5, lines 42-46, page 6, lines 1-4). Revision: Group 1 will include 100 mother-infant dyads with infants that are 1 month \pm 30 days of age, Group 2 will include 100 mother-infant dyads with infants that are 3 months \pm 30 days of age, and Group 3 will include 50 mother-infant dyads with infants that are 6 months \pm 30 days of age. As neuroimaging scans in natural sleep are more likely to be successful in

	infants of younger age and MRI scans with sedation may be more challenging to obtain enrollments for, the group sizes of the younger groups (1-month and 3-month) are larger than that of the older infants (6-months). This split will also allow the team to obtain images and samples at different timepoints during the early childhood period without the need for repetitive sampling of any one mother-child dyad.
page 6 line 40-41, 50-53; it would be good to mention the frequency of these visits and the MRI scans in the methodology as opposed to leaving that in the figure alone	This is included in the text (page 6, lines 6-8), and more in-depth in the Measures of outcomes section (pages 6-8).
page 6 line 48; Elaborate on which quality control measures have been put in place for data collection as mentioned, to ensure there are no changes in the measurement of variables over the duration of the study.	This has been updated (page 6, lines 15-22). Revision: Additionally, quality control programs are in place for data collection to ensure there are no changes in the measurement of variables over the study's duration. Weekly quality checks will be conducted under the supervision of master trainers for all behavioral, cognitive, and developmental assessment tools used in the study. Routine quality assurance of scans will be performed by MRI technicians immediately after the scan is obtained and again prior to image preprocessing. The quality of biological samples will be checked by the lab team and frequent quality management of all collected data will be done by the Data Management Unit.
The information in the figure given should match with the methodology as much as possible, especially in key areas. Example, visit 2 in the figure states demographic information alone yet the methodology states demographic and socioeconomic data (page 7 line 19)	The information in the figure has been updated to match that in the text (Figure 1).
Maternal factors in page 7 line 36-39 are specified for assessment at visit 1. However, maternal factors like anthropometry and behavioral and cognitive (page 8 line 9) outcomes can change at any point of a child's growth, especially in regard to the ages included in the study during follow up, thus affecting neurodevelopment. These maternal factors could be assessed at every visit.	This has been updated so that these measures are collected at "all visits" (page 7, line 2, 13, 24).
Child's history, page 7 line 27 is	This has been updated so that this measure is collected at "all

specified at the first year, yet this history is contributory at every stage of the early child development. Major hospitalizations due to CNS or severe infections, for example, at any point in the first year or two of life can affect an otherwise previously well child's neurodevelopment and MRI finding. It is recommended that this history be taken at every visit.	visits" (page 6, line 40).
page 8 line 23-28; Specify ages in which MUAC will be conducted as opposed to all the children stated in the work.	MUAC will be conducted at each visit for children in each enrollment group: 1, 3, and 6 months; 1 year 1 month, 1 year 3 months, and 1 year 6 months; 2 years 1 month, 2 years 3 months, 2 years 6 months; 3 years 1 month, 3 years 3 months, 3 years 6 months. This has been updated in the text to be clearer (page 7, lines 34-35). Revision: Mid-upper arm circumference will also be measured for children in each enrollment group at every visit.
page 8 line 23-28 specify which growth curve will be used for anthropometric interpretation.	The W.H.O. Z-scores will be used for anthropometric interpretation. This has been updated (page 7, lines 35-39). Revision: The WHO Z-scores will be used for anthropometric interpretation and standards for weight, length/height, weight-for-length/height, and head circumference-for-age will be calculated. Child anthropometric measurements will be collected yearly (visits 1-4) and analyzed within their enrollment group and visit number.

VERSION 2 – REVIEW

REVIEWER	Kim Cecil University of Cincinnati College of Medicine
REVIEW RETURNED	20-Apr-2023

GENERAL COMMENTS	Given the importance of the imaging in this work, more imaging information is requested with this review for the manuscript. What type of Toshiba 3 Tesla MRI scanner is being used for the study? What type of head coil is being used (number of channels, basic design). What is the gradient performance, software platform, etc. MRI Protocol concerns remain with the revision. 1. Common MRI sequence details including type of sequence, key parameters, resolution of the sequence acquisition, etc., remain
--

absent. These can be presented in a table or text. Please define any abbreviations used (for example, co for coronal, sg or sag for sagittal, ax for axial, DWI for diffusion weighted imaging, FSBB??).

2. The selections and prioritization of sequences shown in the protocol (scan card-Figure 2) are concerning. Shouldn't the key sequences for the research objective of longitudinally mapping trajectories of child growth and neurodevelopment be performed first? These would include the 3 dimensional sagittal T1 weighted and T2 weighted volumetric sequences, the DTI and the fMRI? The purpose of the other listed sequences is uncertain.

3. The duration of the fMRI seems excessively short; however, without details it is impossible to know the resolution, use of multi band or multi echo sequences, etc. It will not have sufficient temporal resolution for meaningful analyses.

4. Why is a contrast administration sequence used for this research? (Last listed as 3D T1 wi sg +C)? Unless there is some justification that malnutrition and other environmental factors under study need delineation with contrast administration, there are ethical concerns for its (single as well as the repeated annual) usage in a setting without a clinical indication.

5. What is the purpose of the Coronal FLAIR sequence in this study population?

6. Greater specific detail should be provided for the rationale for the selected sequences, the plan for post-processing each sequence and intended specific analyses. Collecting data that will not address the study objectives wastes resource time (scanner and staff) and potentially lead to participant attrition with excessively long examinations.

7. The manuscript states several times that the total scan time is 32:27 minutes. This is not accurate. The scan protocol requires a minimum of 32:27 minutes for completion. This does not indicate the prescan period for optimizing the scanner for the sequence acquisitions. There should be stated some expected total scan duration that accounts for the scan protocol, the prescanning, the time for getting the child comfortable in the scanner, time for repeating due to motion, etc. As an example, "we anticipate participants to be in the scanner environment no longer than 2 hours." Please rephrase the comments, especially upon consideration of the protocol as requested above and the reality that is occurring with the study, assuming a start date of May 2022.

8. The comments, "Routine quality assurance of scans will be performed by MRI technicians immediately after the scan is obtained and again prior to image preprocessing" need revision. There should be one statement about quality assurance associated with technical performance of the scanner. It may include how often the vendor service engineer tests the system. Also, how often the technologist runs a phantom (daily, weekly) that is analyzed for issues with the system performance? There is also a distinct process associated with assessing quality of an individual sequence performance with participants. This quality assurance occurs for all sequences that are analyzed in group analyses, such as the 3D anatomical, fMRI and Diffusion sequences.

Other concerns. Why did the authors change the number of participants within each group? Wouldn't they need to change the ethics approval by their university board. Given the intended start date listed as May 2022, the changes within the revision need to be discussed.

REVIEWER	Peter Etim Ekanem Mekelle University, Human Anatomy
REVIEW RETURNED	20-Apr-2023

GENERAL COMMENTS	1. The title now suggested is not within the scope of the work as what the study is about The study is not about "METHODS" to image and assess the neurodevelopmental impact of environment in early years, rather it seeks to assess the neurodevelopmental impact of the environment on the early years. Please revise. Consider: "Maternal and Environmental Impact Assessment on Neurodevelopment in Early Childhood Years: A Prospective Cohort Study Protocol from a Low, Middle- Income Country" if acronym (MINE) is what the authors seek to maintain. 2. Please specify which stakeholders and policy makers you are referring to in page 10, lines 27-30. 3. Page 5 line 13 Consider excluding the use of the word "All" because you are not including all. That is why you have exclusion criteria. 4. Page 5 line 16. If the conditions in parentheses are not definitive of hypoxic ischemic encephalopathy, consider removal of the parentheses so that each factor stands alone. 5. Page 5 lines 32-36. The explanation given does not qualify simple random sampling if all the infants are being screened for eligibility. The choice and approach mentioned is what should be further explained to demonstrate how the simple random sampling will be conducted if this study can be replicated. 6. Page 5 lines 42-46, Please state the rationale for widening the age bracket allowance. 1 month with a 30 day allowance would mean a neonate is included and a 2 month old covered by the second group and a 5 month old by the third class. Your classification therefore covers a newborn, one month old, 2 month old, 5, and 6 month old. Is that right? This also does not then tally with the inclusion criteria. 7. Page 6 line 6-8. Instead of stating four visits, kindly specify the frequency. They are yearly visits or once a year? 8. Page 7 line 33. Is MUAC used in newborns? Please specify MUAC as per age and length/height criteria for use of MUAC for anthropometry.
---

VERSION 2 – AUTHOR RESPONSE

Reviewer 1: Prof. Kim Cecil	Response
What type of Toshiba 3 Tesla MRI scanner is being used for the study? What type of head coil is being used (number of channels, basic design). What is the gradient performance, software platform, etc.	We have updated the scanner information to be more detailed. Revision: Page 8: At every visit (1-4), MRI scans will be conducted using a 3T MRI machine (Toshiba Titan, Canon Medical Systems) equipped with a 32-channel coil, which provides high field strength, multi-phase imaging, and advanced motion correction capabilities. To optimize the functionality and performance of the Titan scanner, the specialized Vantage 2.5 software

	will be utilized.
Common MRI sequence details including type of sequence, key parameters, resolution of the sequence acquisition, etc., remain absent. These can be presented in a table or text. Please define any abbreviations used (for example, co for coronal, sg or sag for sagittal, ax for axial, DWI for diffusion weighted imaging, FSBB??).	The MRI sequence details and abbreviations have been updated in the text and in the table (Table 1) and additional information about the sequences performed but not used for analysis are included in the Supplementary Information. Both tables are also included at the end of this document for your reference.
The selections and prioritization of sequences shown in the protocol (scan card-Figure 2) are concerning. Shouldn't the key sequences for the research objective of longitudinally mapping trajectories of child growth and neurodevelopment be performed first? These would include the 3-dimensional sagittal T1 weighted and T2 weighted volumetric sequences, the DTI and the fMRI? The purpose of the other listed sequences is uncertain.	The scanning protocol was developed through the hospital and funding agency's advice. All listed sequences are necessary to complete for our study/hospital agreement and we have been able to successfully obtain all sequences in our participants so far. We have added this into our protocol on page 8: As part of the hospital protocol, Fluid attenuated inversion recovery (FLAIR) and Flow Sensitive Black Blood (FSBB) susceptibility-weighted imaging (SWI) will also be performed and reviewed by a physician, but these sequences will solely be used for clinical information.
The duration of the fMRI seems excessively short; however, without details it is impossible to know the resolution, use of multi band or multi echo sequences, etc. It will not have sufficient temporal resolution for meaningful analyses.	The fMRI scan sequence takes three minutes and 41 seconds, which is shorter than other sequences, but enables faster imaging and decent temporal resolution. The parameters are as follows TR: 2400, TE: 25, FOV: 240x240, Matrix: 96x96, Voxel size: 2.5x2.5x4 We also intend to use a variety of preprocessing techniques to better the SNR for analyses.
Why is a contrast administration sequence used for this research? (Last listed as 3D T1 wi sg +C)? Unless there is some justification that malnutrition and other environmental factors under study need delineation with contrast administration, there are ethical concerns for its (single as well as the repeated annual) usage in a setting without a clinical indication.	Thank you for pointing this out. We are not using a contrast in this study and the sequence name was mislabeled in the scanner, but this has been corrected.
What is the purpose of the Coronal FLAIR sequence in this study population?	As the FLAIR sequence is part of the hospital's protocol, we must include this sequence in our study's scanning.

Greater specific detail should be provided for the rationale for the selected sequences, the plan for post-processing each sequence and intended specific analyses. Collecting data that will not address the study objectives wastes resource time (scanner and staff) and potentially lead to participant attrition with excessively long examinations.	We have provided details related to the sequences we will be analyzing and how. As this is the first time our university will be conducting this type of analysis, our funding agency has been advising us on various analysis tools through the Flywheel platform that we will be using for analysis (e.g., FreeSurfer and FSL for structural/functional, Mrtrix3 for diffusion). We have also added a statement that explains why we are collecting some measures that will not be analyzed (they will only be used for clinical information). Addition: Page 8: As part of the hospital protocol, Fluid attenuated inversion recovery (FLAIR) and Flow Sensitive Black Blood (FSBB) susceptibility-weighted imaging (SWI) will also be performed and reviewed by a physician, but these sequences will solely be used for clinical information.
The manuscript states several times that the total scan time is 32:27 minutes. This is not accurate. The scan protocol requires a minimum of 32:27 minutes for completion. This does not indicate the prescan period for optimizing the scanner for the sequence acquisitions. There should be stated some expected total scan duration that accounts for the scan protocol, the prescanning, the time for getting the child comfortable in the scanner, time for repeating due to motion, etc. As an example, "we anticipate participants to be in the scanner environment no longer than 2 hours." Please rephrase the comments, especially upon consideration of the protocol as requested above and the reality that is occurring with the study, assuming a start date of May 2022.	This change has been updated. Since the hospital team only allots one hour for each MRI scan (prescan to completion), we complete our scans within that time frame. Revision: Page 6: Once the scan is completed, estimated to take no longer than 1 hour (prescanning (5 minutes), child placement (10 minutes), scanning time (30+ minutes, depending on repeated measures))... Page 8: At every visit (1-4), MRI scans (lasting no longer than one hour) will be conducted using a 3T MRI machine... Page 9: Though the total scan time is 32:27 minutes, repeating sequences depending on motion artifacts may result in the scan lasting longer. Page 9: The sequences affected by participant motion will be repeated once the child is asleep again.
The comments, "Routine quality assurance of scans will be performed by MRI technicians immediately after the scan is obtained and again prior to image preprocessing" need revision. There should be one statement about quality assurance associated with technical performance of the scanner. It may include how often the	The quality assurance related to MRI has been consolidated to page 6 where the other quality assurance processes are listed. Information related to assessing the quality of the scans has been moved to page 9 in the statistical analysis section.

vendor service engineer tests the system. Also, how often the technologist runs a phantom (daily, weekly) that is analyzed for issues with the system performance? There is also a distinct process associated with assessing quality of an individual sequence performance with participants. This quality assurance occurs for all sequences that are analyzed in group analyses, such as the 3D anatomical, fMRI and Diffusion sequences.	Revision: Page 6: For the neuroimaging data, to ensure the scanner is operating correctly and producing reliable images, radiologists conduct routine quality control and assurance procedures weekly, such as system checks and calibrations (with a phantom), under the supervision of the biomedical team. In addition, a service engineer performs thorough diagnostic/maintenance tests on a quarterly basis to confirm the machine is working as intended. If there are any machine or software malfunctions, scans will not be completed during that time. Page 9: While the length of the MRI exam will be affected by various factors, including patient motion and machine problems or troubleshooting during the scan (i.e. scanner calibration), all MRI data will undergo routine quality assurance. MRI data will be checked immediately after acquisition to ensure 1) no abnormalities were found, 2) all sequences were completed, and 3) there was limited motion. The MRI machine will also be continuously checked to ensure it is running properly, and if there are any machine or software malfunctions, scans will not be completed during that time. Page 9: MRI data will be quality checked by a radiologist and radiologist technician to ensure there were no abnormalities found, all sequences were completed, and there was limited motion. Visualization of data will be done using Mango and Horos tools.
Why did the authors change the number of participants within each group? Wouldn't they need to change the ethics approval by their university board. Given the intended start date listed as May 2022, the changes within the revision need to be discussed.	During the pilot testing, initial MRI scans for 6-month-olds were unsuccessful in natural sleep, and when we were approved to complete these scans with sedation, the refusal rate was a lot higher than for the 1-month and 3-month-old infant scans in natural sleep. For this reason, we reached out to our funding agency for advice, and they advised us to change the number of participants in each group. We submitted this change, as well as any other indicated changes, to our Ethics Review Committee and only implemented changes after approval.
Reviewer 2: Dr. Peter Etim Ekanem	Response
The title now suggested is not within the scope of the work as what the study is about. The study is not about "METHODS" to image and assess the neurodevelopmental impact of environment in early years, rather	Thank you for this suggestion, especially as it maintains the MINE acronym which we have been utilizing in our study. We have updated our study title to your suggestion: Maternal and Environmental Impact Assessment on Neurodevelopment in Early

it seeks to assess the neurodevelopmental impact of the environment on the early years. Please revise. Consider: “Maternal and Environmental Impact Assessment on Neurodevelopment in Early Childhood Years: A Prospective Cohort Study Protocol from a Low, Middle- Income Country” if acronym (MINE) is what the authors seek to maintain.	Childhood Years: A Prospective Cohort Study Protocol from a Low, Middle- Income Country.
Please specify which stakeholders and policy makers you are referring to in page 10, lines 27-30.	This has been updated. Revision: As this study may provide insight to factors in the environment that will influence child development, the key points from the study will be shared with stakeholders and policymakers (e.g., Pakistan Pediatrics Association, Maternal and Child health, Government of Sindh), who can use the study’s findings to plan interventions for the physical and mental health of mothers and children at national, regional and international levels.
Page 5 line 13 Consider excluding the use of the word “All” because you are not including all. That is why you have exclusion criteria.	We have removed the word “All”. Revision: Singleton infants
Page 5 line 16. If the conditions in parentheses are not definitive of hypoxic ischemic encephalopathy, consider removal of the parentheses so that each factor stands alone.	This has been updated. Revision: Infants with history of hypoxic-ischemic insult, delayed cry, requiring oxygen, or hospitalization
Page 5 lines 32-36. The explanation given does not qualify simple random sampling if all the infants are being screened for eligibility. The choice and approach mentioned is what should be further explained to demonstrate how the simple random sampling will be conducted if this study can be replicated.	The list of newborns shared from the PRiSMA team is only screened to identify infants that fit the age criteria. Then, all houses with infants of the appropriate age are randomly approached. In the home, in-depth medical history of the mother and child are collected to determine eligibility (based on exclusion criteria). This has been updated: Page 5: A list of households with newborns will be

	shared by the PRiSMA study team on a monthly basis. After households with infants of appropriate age are identified, mother-child dyads will be approached via simple random sampling to reduce any sampling bias. At the mother's home, research staff will ask questions related to their and their child's medical history to determine their eligibility (based on the exclusion criteria).
Page 5 lines 42-46, Please state the rationale for widening the age bracket allowance. 1 month with a 30 day allowance would mean a neonate is included and a 2 month old covered by the second group and a 5 month old by the third class. Your classification therefore covers a newborn, one month old, 2 month old, 5, and 6 month old. Is that right? This also does not then tally with the inclusion criteria.	The age bracket allowance was widened due to a variety of reasons which prevented infant scans from occurring when the child was exactly X# months. Logistically, with the difficulties in obtaining consent from the family in a timely manner (since family members are not always at home and most do not have mobile numbers to call), scheduling issues related to child's health, family availability, and scanner availability (as we are utilizing one in a hospital facility with a large patient load), a narrow age bracket allowance prevented us from recruiting many families in our study. Additionally, some deliveries would not be reported in a timely manner to the surveillance site, so the child's age would sometimes be available later. With the younger infants, a cultural issue that affected recruitment was that in many areas, women are not allowed to leave their homes for the first month after delivery. As there were many challenges associated with recruiting infants within a narrow bracket, the expanded bracket allows us to more easily recruit infants 1-2 months old, 3-4 months old, and 6-7 months old. We have updated the inclusion criteria to reflect this. Revision: Singleton infants aged 1 month \pm 30 days of age, 3 months \pm 30 days of age, or 6 months \pm 30 days of age of age and their mothers
Page 6 line 6-8. Instead of stating four visits, kindly specify the frequency. They are yearly visits or once a year?	This has been updated to be more specific. Revision: The study involves the mother-child dyads completing

	four yearly visits, beginning when the child is an infant (1 month/3 months/6 months) until they are 3 years and 1 month/3 months/6 months of age.
Page 7 line 33. Is MUAC used in newborns? Please specify MUAC as per age and length/height criteria for use of MUAC for anthropometry.	We will only use MUAC for children above six months of age. For infants less than six months of age, we will only use the WHO weight for age and weight for length Z scores. Revision: Page 7: Mid-upper arm circumference will also be measured for children above six months of age at every visit.

VERSION 3 – REVIEW

REVIEWER	Kim Cecil University of Cincinnati College of Medicine
REVIEW RETURNED	24-May-2023

GENERAL COMMENTS	The authors have addressed the major concerns from the prior critiques.
---